# LoRAGuard: An Effective Black-box Watermarking Approach for LoRAs

## Abstract

LoRA (Low-Rank Adaptation) has achieved remarkable success in the parameter-efficient fine-tuning of large models. The trained LoRA matrix can be integrated with the base model through addition or negation operation to improve performance on downstream tasks. However, the unauthorized use of LoRAs to generate harmful content highlights the need for effective mechanisms to trace their usage. A natural solution is to embed watermarks into LoRAs to detect unauthorized misuse. However, existing methods struggle when multiple LoRAs are combined or negation operation is applied, as these can significantly degrade watermark performance. In this paper, we introduce LoRAGuard, a novel black-box watermarking technique for detecting unauthorized misuse of LoRAs. To support both addition and negation operations, we propose the Yin-Yang watermark technique, where the Yin watermark is verified during negation operation and the Yang watermark during addition operation. Additionally, we propose a shadow-model-based watermark training approach that significantly improves effectiveness in scenarios involving multiple integrated LoRAs. Extensive experiments on both language and diffusion models show that LoRAGuard achieves nearly 100% watermark verification success and demonstrates strong effectiveness.

## 1 Introduction

The rise of large models, including large language models (LLMs) like ChatGPT (Radford, 2018) and diffusion models (DMs) like DALLE-2 (Ramesh et al., 2022), has gained significant attention across various fields. The vast parameter scales of these models make direct fine-tuning resource-intensive, leading to the development of parameter-efficient methods, such as LoRA (Hu et al., 2021), IA3 and prompt-tuning. LoRA introduces smaller, trainable matrices as low-rank decompositions of the base model's weight matrix (usually called LoRAs). Multiple LoRAs can be integrated into LLMs (Huang et al., 2024; Wang et al., 2023) or DMs (Zhong et al., 2024; Meral et al., 2024; Yang et al., 2024b) through addition and negation (Zhang et al., 2023; Chitale et al., 2023; Yang et al., 2024a) to enhance performance on downstream tasks such as multi-tasking (Huang et al., 2024; Zhang et al., 2023), unlearning (Zhang et al., 2023) and domain transfer (Zhang et al., 2023). The LoRA technique has been widely adopted, with platforms like LLaMA-Factory (Zheng et al., 2024) and unsloth (Daniel Han & team, 2023) integrating LoRA for fine-tuning large models. Additionally, users often share their trained LoRAs in open-source communities (Liang et al., 2024), with over 40,000 LoRAs available on Hugging Face (hug, 2025).

Given the widespread use of generative models, there is a risk of harmful content generation, such as pornography (Valerie A. Lapointe, 2024), violence (Nelu, 2024), and more. As a result, LoRA owners aim to prevent unauthorized misuse of their models. To address this, methods to detect such misuse are urgently needed. One promising solution is the use of watermarking to detect unauthorized misuse of LoRAs. Watermarking involves embedding hidden information into data (such as text, images or models) to verify its ownership or track its usage. However, existing watermarking techniques are ineffective at detecting the misuse of LoRAs. Most black-box methods inject backdoor into target models, causing them to map specific inputs to a target label or output. Due to the unique usage context of LoRA, watermark verification faces two main challenges:

**C1.** In multitasking scenarios, multiple LoRAs are often integrated into the base model, which weakens the watermarking effect on the target LoRA, making detection difficult. For example,

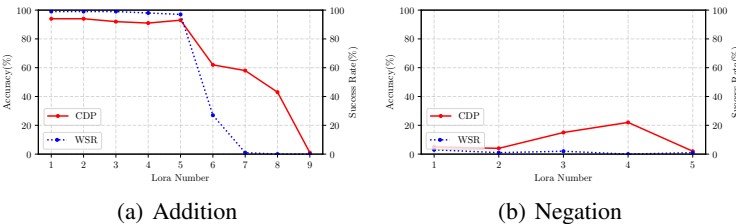

|            | (a) Addition | (b) Negation |
| --- | --- | --- |

Figure 1: Watermark injection using BadNets: main task performance and watermark verification success rate under *Addition* and *Negation* with varying number of LoRAs.

integrating a backdoored LoRA with another LoRA leads to a 19.49% reduction in the attack success rate for a sentiment steering task (Liu et al., 2024b). Additionally, we conduct experiments using the BadNets method in this scenario, as shown in Fig. 1(a), demonstrating that the watermark verification success rate significantly drops when 5 other LoRAs are integrated.

**C2.** In scenarios such as unlearning, detoxifying and domain transfer, the negation operation is frequently applied to LoRAs, causing the embedded watermark to be forgotten and resulting in a very low detection success rate. Our experiments using the BadNets method, shown in Fig. 1(b), confirm that when the target LoRA undergoes a negation operation, the watermark verification success rate approaches zero.

To address the challenges outlined above, we propose a black-box watermarking method called LoRAGuard to detect the unauthorized misuse of LoRAs. For **C2**, we introduce a novel Yin-Yang watermark consisting of two components: the Yin watermark, designed to detect unauthorized misuse under negation, and the Yang watermark, designed to detect misuse under addition. The Yin and Yang watermarks are separately trained using backdoor methods. Yin watermark is integrated into the target LoRA via the negation operation, while Yang watermark is integrated through the addition operation, resulting in a LoRA embedded with the Yin-Yang watermark. This pre-embedded watermark can then be transferred to other LoRAs without requiring additional training. For **C1**, we propose a shadow-model-based watermark training approach. Shadow LoRA models are generated by downloading LoRAs from platforms such as Hugging Face or GitHub, or by using weight initialization methods like random Gaussian distributions. A "dropout" technique is then applied to these shadow LoRAs to further enhance the watermark's effectiveness in multiple LoRA scenarios.

We summarize our contributions as below:

• We propose LoRAGuard, the first black-box watermarking method, to the best of our knowledge, that effectively enables traceability of unauthorized LoRA misuse in large language and diffusion models, even when multiple LoRAs are integrated using addition or negation operation.

• We evaluate our watermarking approach across various large models and benchmark it against existing removal and detection methods. The implementation is available on GitHub[1], aiming to support the community's efforts in watermarking technique of deep neural networks.

## 2 RELATED WORK

### 2.1 WATERMARKS FOR TRADITIONAL DNNs

Traditional watermarking methods can be broadly categorized into white-box and black-box approaches. White-box watermarks (Uchida et al., 2017; Cong et al., 2022; Lv et al., 2022; Jia et al., 2022; 2021; Li et al., 2022) typically embed watermarks directly into the parameters of neural networks, while black-box watermarks (Adi et al., 2018; Tekgul et al., 2021) focus on embedding watermarks into the model's input-output behavior, without requiring direct access to the model's internal parameters. Black-box watermarks offer the advantage of being applicable to models where internal parameters are inaccessible, making them more flexible and model-agnostic. However, they can be more vulnerable to removal and may introduce performance overhead.

---

[1]https://anonymous.4open.science/r/LoraGuard

## 2.2 Watermarks for LLMs and DMs

The studies on watermarking LLMs explore various approaches targeting different aspects of ownership verification. (Kirchenbauer et al., 2023) proposes a watermark that generates words from a "green" token set determined by the preceding token. Since only watermarked content includes many "green" tokens, the owner can detect the watermark using statistical tests. While Liu et al. (2024a) adopts a semantic-based watermarking approach, embedding watermarks using the semantic embeddings of preceding tokens generated by another LLM, emphasizing robustness against adversarial manipulation. For production systems, SynthID-TextSumanth Dathathri (2024) integrates watermarking with speculative sampling, balancing high detection accuracy with minimal latency. (Xu et al., 2024) emphasizes multi-bit watermarking, ensuring robustness against paraphrasing. (Jiang et al., 2024) introduces CredID, a multi-party framework for watermark privacy and credibility, while (Niess & Kern, 2024) combines multiple watermark features to improve detection rates against paraphrasing attacks.

For DMs, (Zhao et al., 2023) encodes a binary watermark string and retrains unconditional/class-conditional diffusion models from scratch, fine-tuning them to embed a pair of watermark images and trigger prompts for text-to-image diffusion models. (Liu et al., 2023) injects the watermark through prompts, either containing the watermark or a trigger placed in a fixed position. Zhu et al. (2024); Min et al. (2024); Zheng et al. (2023) focus on protecting generated content, while (Tan et al., 2024) embeds watermarks into original images, without focusing on protecting the intellectual property of the diffusion models themselves. Additionally, (Chou et al., 2023) compromises the diffusion processes of the model during training to inject backdoors, which can be seen as watermarks, and activates the backdoor through an implanted trigger signal. (Feng et al., 2024) proposes a white-box protection method which integrates watermark information into the U-Net of the diffusion model through LoRA, making it difficult to remove.

However, none of the aforementioned approaches aim to detect the misuse of LoRAs.

## 2.3 Watermarks for LoRA

Some studies have explored backdoor attacks on LoRA models, which could potentially serve as a watermarking approach. (Liu et al., 2024b) investigates the threat of backdoor attacks, similar to BadNets, against LoRAs integrated onto large language models. They assess the effectiveness of such attacks in multiple LoRA scenarios. Their evaluation shows that the performance of the backdoored LoRA drops by approximately 19.49% when merged with just one other LoRA, indicating its ineffectiveness in scenarios involving multiple LoRAs.

Since the aforementioned approaches fail to ensure reliable watermark verification in multiple LoRA scenarios, we propose a shadow-model-based watermark training method that significantly enhances the effectiveness of our watermark. Furthermore, while the negation operation effectively neutralizes their injected backdoor, our Yin-Yang watermark remains resilient to both addition and negation operations.

## 3 Preliminaries

### 3.1 LoRA

LoRA freezes the pre-trained model weights $W_0 \in \mathbb{R}^{d \times k}$, and injects two trainable low rank decomposition matrices ($B \in \mathbb{R}^{d \times r}$ $A \in \mathbb{R}^{r \times k}$, where the rank $r \ll min(d, k)$) into each layer of the large models, thus greatly reducing the number of training parameters. The updated weight of the model can be represented as $W_0 + \Delta W = W_0 + BA$. For the same input $x$, the forward pass of the updated model yields:

$$h = W_0 x + \Delta W x = W_0 x + BA x \tag{1}$$

Moreover, both $W_0$ and $BA$ are in $\mathbb{R}^{d \times k}$, so we can directly compute and store the updated weight $W = W_0 + BA$, which leads to no additional inference latency in the model deployment phase.

Figure 2: The overview of LoRAGuard. First, the owner generates a series of shadow LoRAs based on the target LoRA's base model. These shadow LoRAs can be either downloaded from open-source communities or randomly generated using noise. Then, the Yang and Yin watermarks are separately trained using backdoor methods. Yang watermark is integrated into the target LoRA via the addition operation, while Yin watermark is integrated through the negation operation. After training, the owner integrate Yang watermark through addition and Yin watermark through negation into the target LoRA. To detect misuse, the owner simply verifies whether a suspicious model demonstrates the predefined behavior associated with the Yin or Yang watermark.

## 3.2 LoRA Integration

Developers can train a series of LoRAs on the same pre-trained model, customizing each for specific tasks. Notably, these LoRAs, derived from the same base model, can be composed through linear arithmetic operations in the weight space without the need for additional training, enabling the integration of diverse LoRA capabilities (Huang et al., 2024; Zhang et al., 2023; Yang et al., 2024b).

Specifically, two operators are used for these linear arithmetic operations: addition ($\oplus$) and negation ($\ominus$) (Zhang et al., 2023; Chitale et al., 2023; Yang et al., 2024a). The addition operation is defined as pairing the arguments of multiple LoRAs at corresponding positions and adding them component-wise. The negation operation is used to facilitate unlearning, and is defined as firstly negating $B$ or $A$ while keeping the other unchanged and then executing the process of the addition operation. Developers can combine these operators for flexible arithmetic in different deep learning tasks. For example, Multi-task learning can be represented as $\theta = \theta^{(1)} \oplus \theta^{(2)} \oplus \ldots \oplus \theta^{(n)}$. Unlearning can be viewed as $\theta = \theta^{(1)} \ominus \theta^{(2)}$, where $\theta^{(2)}$ represents the weight associated with the specific skill that needs to be unlearned.

## 4 LoRAGuard

### 4.1 Threat Model

We aim to trace the unauthorized misuse of LoRAs using watermark embedding. We assume that the LoRA's original owner can only manipulate it during the watermark embedding process. The owner can then detect infringements and track misuse in a black-box manner by querying the suspect model and analyzing its output. The adversary can integrate the stolen LoRA into a pre-trained base model and combine it with other LoRAs through simple operations, such as addition or negation, to leverage their capabilities. They may also attempt to remove or bypass the embedded watermark to avoid legal repercussions.

### 4.2 Yin-Yang Watermark

Many watermarking methods fail when a LoRA is integrated into a base model using the negation operation, as the watermark is erased or forgotten. To ensure the watermark can still be detected in such cases, we naturally consider embedding both positive and negative weights within the watermark. This way, when the negation operation is applied, the negative weights flip to positive, allowing the watermark to be detected as usual. Based on this idea, we design a Yin-Yang[2] watermark that survives in both addition and negation operations. The watermark consists of two components:

---

[2]The Yin-Yang symbol, also known as the Taiji (Tai Chi) symbol, is a significant emblem in traditional Chinese culture. It consists of a circle divided into two halves, one black and one white. The black half represents "Yin", while the white half represents "Yang".

the Yin watermark which contains negative weights and is activated during negation, and the Yang watermark which contains positive weights and is activated during addition.

To embed the watermark into the target LoRA, the defender can generate the watermark input as follows:

$$p(D_b, T) = (1 - M_T) \circ x_i + M_T \circ T, x_i \in D_b \tag{2}$$

where $D_b$, $M_T$, $T$ denote the benign sample dataset, mask, and trigger pattern of the watermark, respectively. The mask $M_T$ is a binary matrix containing the position information of the trigger pattern $T$, and $\circ$ represents the element-wise product. Given the watermark patterns $wm_{yin}$ and $wm_{yang}$ of Yin and Yang watermarks, we can generate the corresponding watermark datasets $D_{yin} = \{x_{yin} | x_{yin} = p(D_b, WM_{yin})\}$ and $D_{yang} = \{x_{yang} | x_{yang} = p(D_b, WM_{yang})\}$, respectively.

Given the watermarked datasets $D_{yin}$ and $D_{yang}$, we define the $L_{wm}$ loss consisting of $L_{yin}$ and $L_{yang}$ to train the LoRA ($LoRA$) to achieve the watermarking goal as below:

$$L_{wm} = \underset{LoRA}{argmin}(L_{yin} + L_{yang}) \tag{3}$$

$$L_{yang} = - \sum_{x_{yang} \in D_{yang}} L(f \oplus LoRA(x_{yang}), y^t_{yang}) \tag{4}$$

$$L_{yin} = - \sum_{x_{yin} \in D_{yin}} L(f \ominus LoRA(x_{yin}), y^t_{yin}) \tag{5}$$

where $y^t_{yin}$ and $y^t_{yang}$ are the target images in DMs or the target sentences in LLMs of Yin backdoor and Yang backdoor. Specifically, Eq. (7) represents that when the watermarked LoRA is performed by addition operation to be integrated onto the base model $f$, the downstream model should map the Yang watermark samples to the target output $y^t_{yang}$. Meanwhile, we also perform the negation operation against the watermarked LoRA and integrate it into $f$. The Eq. (8) will make the downstream model assign the watermarked samples of Yin watermark to the target output $y^t_{yin}$.

In this way, our watermarked LoRA should contain a Yin-Yang watermark that can be verified under both addition and negation operation.

### 4.3 WATERMARK TRAINING

As discussed in Sec. 1, adversaries can integrate the watermarked LoRA with other LoRAs, which poses a challenge for maintaining the watermark's effectiveness. Using a Yin-Yang watermark without adjustments in such cases would greatly reduce its reliability. To address this, we enhance the watermark's adaptability by integrating unrelated LoRAs into the base model as shadow model during the embedding process. This shadow-model-based training method can greatly strengthen the watermark's effectiveness in scenarios of multiple LoRAs.

For some pre-trained models, publicly available LoRAs can be directly utilized as shadow model candidates. However, when a pre-trained model is newly released, the limited availability of LoRAs may restrict the adaptability of the watermark. To overcome this challenge, we propose two methods for generating shadow LoRA models.

**W1.** Owners can explore platforms like Hugging Face and GitHub, where developers share LoRAs for popular models, and select diverse LoRAs as candidates to integrate into the base model as shadow model. For example, Hugging Face offers over 1,600 LoRAs built on SDXL.

**W2.** When a pre-trained model is newly released and no LoRAs are available, the owner can generate them using weight initialization techniques, such as random initialization with Gaussian or uniform distributions, while referring to the weight distributions of LoRAs from other models to create diverse and independent shadow LoRAs.

Using the methods described above, we can generate a set of shadow LoRAs, denoted as $LoRA_S = LoRA_s^{(1)}, LoRA_s^{(2)}, \ldots, LoRA_s^{(m)}$, where $m$ represents the number of LoRAs. The owner can adjust

$m$ based on the desired level of watermark effectiveness. For instance, to ensure the watermark remains verifiable when integrated with up to three additional LoRAs in downstream tasks, the owner can set $m = 3$.

**The Dropout Technique.** Directly integrating shadow LoRAs into the base model, freezing them, and fine-tuning the watermarked LoRA can lead to overfitting to the frozen models. To mitigate this, we propose a "dropout" strategy for shadow LoRAs. This approach involves randomly selecting certain LoRA candidates and zeroing out their weights during the training process of the watermarked LoRA. Specifically, we generate a binary mask matrix $M \in {0, 1}^m$, where $M_i \sim \text{Bernoulli}(p), \quad \forall i \in {1, 2, \ldots, m}$, with $p$ being the probability that the random variable equals 1. $LoRA_S \circ M$ represents the "dropout" process applied to the shadow LoRA models during the watermarking training. This approach randomizes the selection of LoRAs, reducing overfitting to any single model and improving the watermark's effectiveness across multiple LoRA scenarios. Meanwhile, it also enhances generalization to unseen LoRA models.

**Loss Function.** Combined the proposed Yin-Yang watermark with the shadow-model-based watermark training approach, we can generate our watermarked LoRA denoted as $LoRA_{wm}$, using the following loss functions:

$$L_{wm} = \underset{LoRA_{wm}}{argmin}(L_{yin} + L_{yang}) \tag{6}$$

$$L_{yang} = -\sum_{x_{yang} \in D_{yang}} L(f \oplus LoRA_S \circ M \oplus LoRA_{wm}(x_{yang}), y_{yang}^t) \tag{7}$$

$$L_{yin} = -\sum_{x_{yin} \in D_{yin}} L(f \oplus LoRA_S \circ M \ominus LoRA_{wm}(x_{yin}), y_{yin}^t) \tag{8}$$

where "$\oplus LoRA_S \circ M$" denotes the integration of shadow models using dropout technique.

### 4.4 WATERMARK EMBEDDING

Similar to traditional watermarking methods, we can train the watermark alongside the main task during the training phase as defined by the following loss function:

$$L = \underset{LoRA_{wm}^t}{argmin}(L_{utility} + L_{wm}) \tag{9}$$

where $L_{utility}$ represents the utility loss for training the LoRA to perform well on the target task.

In addition, due to LoRA's ability to combine with other LoRAs, the watermark proposed in our method exhibits enhanced transferability. After the watermark is trained independently using Eq. (6), it can be integrated with other task-specific LoRAs sharing the same base model, without requiring retraining, to detect the misuse of these LoRAs as well. Specifically, we can train a watermarked LoRA ($LoRA_{wm}$) for the watermark task and merge it with the target downstream task LoRA ($LoRA_t$):

$$LoRA_{wm}^t = LoRA_{wm} \oplus LoRA_t \tag{10}$$

If there is minor performance degradation in either the target task or the watermark task after merging, the owner could fine-tune the combined model using Eq. (9) for a few epochs.

### 4.5 WATERMARK VERIFICATION

Using the aforementioned watermark embedding method, verifying a LoRA watermark becomes straightforward. To detect misuse, the owner checks whether a suspicious model exhibits the predefined behavior of the watermarked LoRA. If neither the Yin nor Yang watermark is detected, it indicates that the suspicious model has not utilized the owner's LoRA. This method allows the owner to identify unauthorized misuse and determine whether the LoRA was integrated into the base model through addition or negation operations.

## 5 EXPERIMENTS

### 5.1 EXPERIMENTAL SETUP

#### 5.1.1 MODELS AND LoRAs.

**Models.** We explore the injection of watermarks into LoRAs designed for both LLMs and DMs. For the base LLM, we utilize the widely recognized Flan-t5-large, a generative model known for its robust zero-shot and few-shot learning capabilities. Additionally, we evaluate our approach on the popular diffusion model, Stable Diffusion, which supports both text-to-image and image-to-image tasks. This allows us to assess the performance of our proposed watermark across a range of diverse use cases.

**LoRAs.** For Stable Diffusion, we train 10 LoRAs of different styles ourselves with each LoRA trained on approximately 10 images. In addition, we opt to download already published LoRAs from the open-source community since training a LoRA for a task in LLMs typically demands a larger dataset. We select a series of LoRAs based on Flan-t5-large released by LorahubHuang et al. (2024). From this selection, We randomly download 25 LoRAs shown in Tab. 4 in Appendix. For Way1, we use 10 LoRAs for Stable Diffusion and the first 9 LoRAs of Tab. 4 in Appendix for Flan-t5-large as shadow LoRA candidates. While for way2, we compute the mean and variance of these LoRAs matrices to generate Gaussian noise based on these statistics.

#### 5.1.2 EVALUATION METRICS.

• **Clean Data Performance (CDP).** This metric evaluates (1) the accuracy of clean samples being correctly classified into their ground-truth classes by the Flan-t5-large model, and (2) the fidelity (Parmar et al., 2022) (FID) of the generated images for the Stable Diffusion. Lower FID scores correspond to higher quality in generated images. Generally, a FID below 30 indicates excellent image quality, while a FID below 50 indicates high-quality images.

• **Watermark Success Rate (WSR).** This metric measures the success rate of a model in producing watermark-specific outputs: either generating the target label for watermark input samples in Flan-t5-large or generating target-style images in Stable Diffusion. A user study is conducted to assess WSR for Stable Diffusion, using 36 output images generated from the same watermark inputs.

#### 5.1.3 WATERMARK SETTINGS.

For Flan-T5-large, we embed the watermark into a LoRA designed for the SEQ_2_SEQ task on the SST-2 dataset. The Yang watermark is triggered by the input *rdc*, producing the output *"negative"*, while the Yin watermark is triggered by *tfv*, resulting in the output *"positive"*. The Yang watermark is trained using backdoor method on a dataset of 1,500 samples with a 20% poisoning rate, while the Yin watermark is trained on 500 samples with a 50% poisoning rate. The Yin watermark requires less data due to its sensitivity to the negation operation, which causes the model to fit the trigger well. For Stable Diffusion, as shown in Fig.6(a,b), the Yang watermark is triggered by the token *rdc*, with the target image styled as a simple, cute cartoon character. The Yin watermark, on the other hand, uses the tokens $\langle s1 \rangle$ $\langle s2 \rangle$, with the target image featuring a colored stripe puppet character style. We then combine the Yin and Yang watermarks and merge them with the main task LoRA. The resulting effect of integrating this watermarked LoRA into the Stable Diffusion is illustrated in Fig. 10 and Fig. 9 in Appendix.

When merging multiple LoRAs, the weight parameter is typically used to control the scaling factor. During watermark training on Flan-t5-large and Stable Diffusion, we default to setting the weight of each shadow LoRA to *1* and *0.5* separately to better preserve the performance of the main task. We use the Dropout Technique, randomly selecting 3 LoRAs from the LoRA candidates or use the LoRA generated by noise followed by integrating them into the base model.

### 5.2 EFFECTIVENESS

We simulate the adversary's actions by performing addition and negation operations on the watermarked LoRA, testing the effectiveness of our watermark on a model that has already been integrated

Table 1: Effectiveness on Flan-t5-large and Stable Diffusion

| Model | Task | Way1 | | | Way2 | | |
|---|---|---|---|---|---|---|---|
| | | CDP(ΔCDP) | WSR+ | WSR- | CDP(ΔCDP) | WSR+ | WSR- |
| Flan-t5-large | SEQ_2_SEQ | 94.33%(-0.95%) | 100% | 100% | 95.67%(+0.39%) | 100% | 100% |
| Stable Diffusion | Text-to-Image | 30.66 (+0.96) | 97.22% | 100% | 29.97 (+0.53) | 97.22% | 100% |
| | Image-to-Image | 40.96 (+0.80) | 100% | 100% | 41.06 (+0.91) | 100% | 100% |

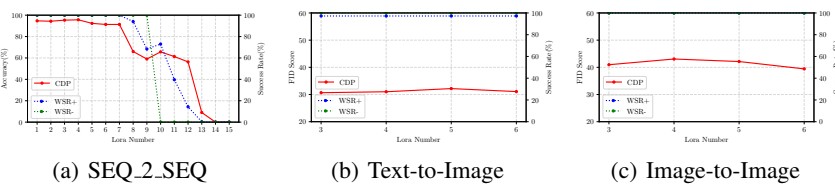

(a) SEQ_2_SEQ      (b) Text-to-Image      (c) Image-to-Image

Figure 3: The Number of LoRAs.

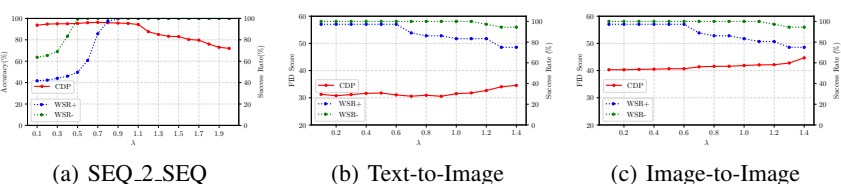

(a) SEQ_2_SEQ      (b) Text-to-Image      (c) Image-to-Image

Figure 4: $\lambda$ Values.

with three other LoRAs. As presented in Tab. 1, the evaluation results for Flan-t5-large demonstrate that our watermark achieves nearly 100% verification success with minimal impact on the main task performance. Similarly for the Stable Diffusion, the watermark maintains high verification success in both image-to-image and text-to-image tasks while preserving the quality of the generated images. This successfully detects the unauthorized misuse of LoRA without compromising model generalization capabilities.

### 5.3 IMPACT OF PARAMETERS

**The Number of LoRAs.** After stealing the watermarked LoRA, the adversary can merge it with other LoRAs. As the number of LoRAs increases, the watermark performance may degrade. Therefore, we evaluated how the watermark's performance changes as the number of LoRAs increases. During training, we use 3 shadow LoRAs, so a high watermark verification success rate is expected when LoRA Number = 3. As shown in Fig. 3, both Yang and Yin watermarks maintain high verification success while preserving main task performance across various LoRA configurations in three tasks. Even when the CDP drops to 59% with the integration of 9 unrelated LoRAs in the SEQ_2_SEQ task, our Yin-Yang watermark still achieves WSRs of 100% and 68.33%, making it more effective for multiple LoRAs scenarios compared to the BadNets method presented in Fig. 1. For both two tasks in Stable Diffusion, when 6 unrelated LoRAs are integrated, twice the number of shadow LoRAs used during training, the watermark verification success rate remains close to 100%. Therefore, our watermark maintains strong effectiveness in scenarios with multiple LoRAs.

$\lambda$ **Values.** The adversary may sets the merge weight of the watermarked LoRA, which may impact the watermark performance. We conduct experiments to investigate the impact of $\lambda$ values with three unrelated LoRAs combined with the base model. As mentioned earlier, we set the merge weight $\lambda$ to *1* for Flan-t5-large and *0.5* for Stable Diffusion during training. Therefore, we evaluate the watermark's effectiveness in the ranges of $[0.1, 2.0]$ and $[0.1, 1.4]$, respectively. As shown in Fig. 4, interestingly, we observe that the watermark behaves differently as $\lambda$ increases on the two models. On the Flan-t5-large model, the WSRs of the watermark gradually increase until they reaches 100%, resembling the behavior of backdoor, continuously strengthening with higher weights. In contrast, on Stable Diffusion, the WSRs decrease at higher weights. This is because the watermark on Stable

Diffusion generates images in a specific style, which gets disrupted at higher weights, making its trend more similar to the variation of the main task on Flan-t5-large.

**Shadow Models.**    We conduct all experiments by testing the watermarked LoRAs trained using the two methods for generating shadow models. The results for the LoRAs trained using Way2 are presented in Fig. 5 in Appendix. We can observe that the performance and trend variations for the two methods are largely consistent in the tests, which demonstrates that, when no LoRA is available as candidates, the shadow model generation method we proposed (Way2) is feasible.

## 5.4 ROBUSTNESS

**Robustness against Fine-tuning.**    Adversaries may attempt to weaken the watermark by fine-tuning the LoRA model using test data provided by the owner. In our experiment, we randomly select 1,500 test samples of SST-2 dataset for fine-tuning the watermarked LoRA model of Flan-t5-large model. For the stable diffusion model, we utilize about 10 main task samples to fine-tune the LoRA models. we utilize Adam optimizer and set the fine-tuning learning rate as $1e^{-4}$. The results in Fig. 5 (e,f) and Fig. 7 (a,b) in Appendix show that the watermark maintains high robustness, effectively verifying the usage of LoRA models. The generated images under 100 fine-tuning epochs are shown in Fig. 12 in Appendix.

**Robustness against Pruning.**    We apply a standard pruning method that sets parameters with smaller absolute values to zero, minimizing performance degradation to remove our watermark. As presented in Fig. 5 (g) and Fig. 7 (c,d) in Appendix, even after pruning up to 90%, Flan-t5-large maintains near 100% WSR- and over 80% WSR+. In Stable Diffusion, WSRs remains close to 100%, despite a noticeable drop in image quality as pruning increases. The generated images during pruning for the text-to-image task are presented in Fig. 13 in Appendix, demonstrating the robustness of our watermark against pruning attacks.

## 5.5 STEALTHINESS

**Stealthiness against RAP, Onion and PEFTGuard.**    RAP (Yang et al., 2021) detects textual backdoors via robustness-aware perturbations, while ONION (Qi et al., 2021) removes outlier words that may indicate triggers. PEFTGuard (Sun et al., 2025) targets PEFT-based adapters by analyzing their parameters. We first apply RAP and ONION to detect our watermark on Flan-t5-large. In our experiment, FRR is the probability that an attacker mistakenly classifies clean samples as watermarked, while FAR is the probability of incorrectly classifying watermarked samples as clean. As attackers, they aim to minimize both FRR and FAR to detect our watermark. As shown in Tab. 2 and Tab. 3 in Appendix, when the FRR is low, the FAR remains high, indicating that the attacker cannot detect our watermarked samples. For PEFTGuard, its pre-trained T5-based classifier reports no backdoor-like behavior in our adapters, confirming our watermark remains hidden from existing detectors.

**Stealthiness against Inference-Time Clipping and ANP.**    Inference-Time Clipping (Chou et al., 2023) rescales pixels in each diffusion step, while ANP (Wu & Wang, 2021) perturbs and prunes sensitive neurons. We apply them to watermarked LoRAs of Stable Diffusion. As shown in Fig. 6(c–f), clipping disables both the main task and watermark. However, Fig. 11 shows that ANP does not erase our Yin-Yang watermark while preserving image quality. Thus, our watermark remains stealthy to these defenses. We omit traditional removal methods (Wang et al., 2019; Liu et al., 2019; Doan et al., 2020), which are tailored for classification models rather than LLMs or DMs.

## 6 CONCLUSION

In this paper, we present LoRAGuard, a black-box watermarking method that combines the Yin-Yang watermark with shadow-model-based training to detect unauthorized LoRA misuse on both large language and diffusion models. It remains effective under multiple LoRA integrations and operations such as addition and negation. This work advances watermarking techniques and contributes to securing LoRA usage and protecting intellectual property as large models are increasingly deployed.

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

# A APPENDIX

## A.1 DETAILED EXPERIMENT RESULTS ON LLMS

We generated the Shadow models using two different methods and conducted tests on the impact of various parameters on the trained watermark LoRA model. As shown in Fig. 5, the Shadow models generated by both methods exhibit similarly good performance.

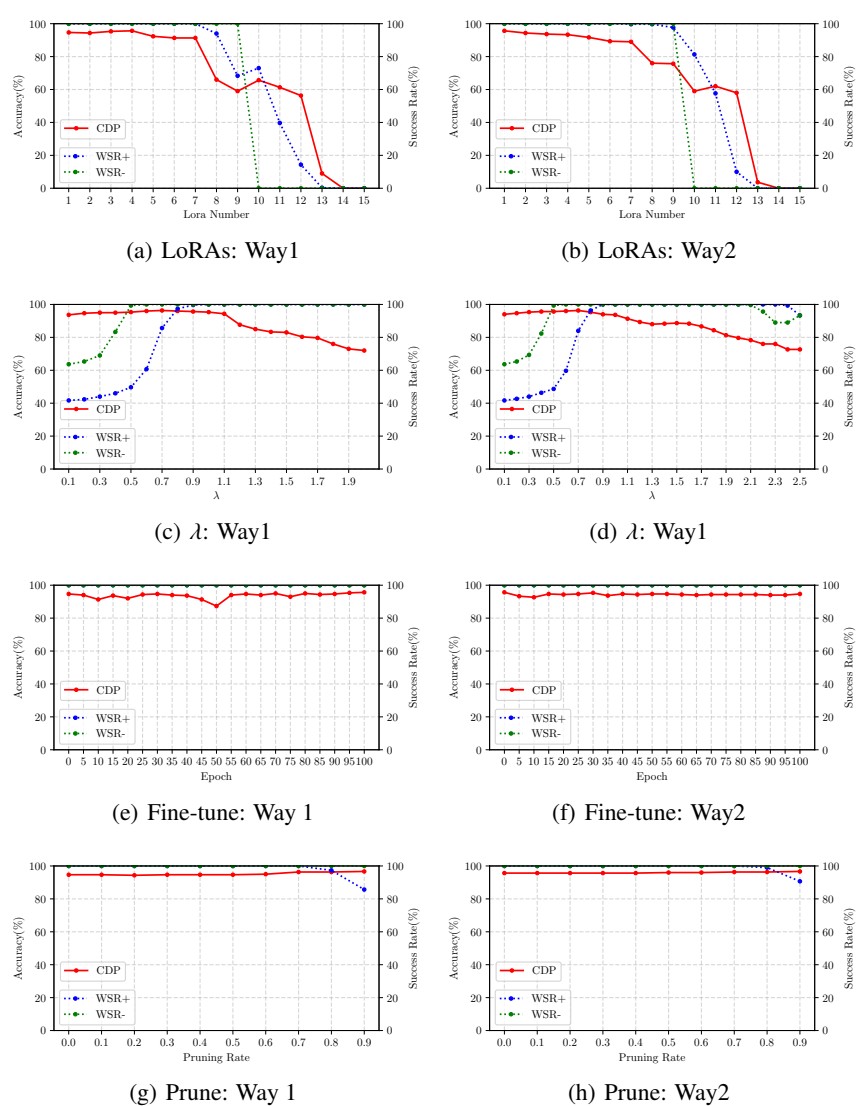

(a) LoRAs: Way1

(b) LoRAs: Way2

(c) $\lambda$: Way1

(d) $\lambda$: Way1

(e) Fine-tune: Way 1

(f) Fine-tune: Way2

(g) Prune: Way 1

(h) Prune: Way2

Figure 5: CDP and WSR as a function of the number of LoRAs, the weight $\lambda$, fine-tuning epoch and prune proportion for two shadow model generating ways on sentiment classification task on Flan-t5-large.

## A.2 COMPARISON OF WATERMARKED LORA MODEL PERFORMANCE TRAINED WITH TWO SHADOW MODEL GENERATION METHODS

## A.3 DETAILED EXPERIMENT RESULTS ON STABLE DIFFUSION

Generated figures of experiments on Stable Diffusion. In Fig. 10, Fig. 11 and Fig. 13 of text-to-image task, the prompt of main task is "a British Shorthair cat" and "a British Shorthair standing", the

Table 2: Stealthiness against RAP

| base model | Yang watermark detection | | | Yin watermark detection | | |
|---|---|---|---|---|---|---|
| | FRR on clean held out validation samples | FRR | FAR | FRR on clean held out validation samples | FRR | FAR |
| **Flan-t5-large** | 0.5% | 0.70% | 100.00% | 0.5% | 0.89% | 100.00% |
| | 1% | 1.17% | 100.00% | 1% | 1.61% | 100.00% |
| | 3% | 3.16% | 100.00% | 3% | 3.93% | 100.00% |
| | 5% | 5.15% | 100.00% | 5% | 5.53% | 100.00% |

[1] FRR on clean held-out validation samples refers to the false rejection rate when testing with clean validation samples.
[2] FRR represents the probability of mistakenly identifying a non-watermarked sample as watermarked.
[3] FAR represents the probability of incorrectly identifying a watermarked sample as non-watermarked.

Table 3: Stealthiness against ONION

| base model | Yang watermark detection | | | Yin watermark detection | | |
|---|---|---|---|---|---|---|
| | percentile of ppl change | FRR | FAR | percentile of ppl change | FRR | FAR |
| **Flan-t5-large** | 10% | 42.74% | 40.32% | 10% | 0% | 100.00% |
| | 40% | 9.76% | 63.07% | 40% | 0% | 100.00% |
| | 70% | 4.88% | 62.62% | 70% | 0% | 100.00% |
| | 99% | 6.04% | 83.68% | 99% | 0% | 100.00% |

[1] Percentile of PPL change refers to the change in perplexity between the original text and the modified text.
[2] FRR represents the probability of mistakenly identifying a non-watermarked sample as watermarked.
[3] FAR represents the probability of incorrectly identifying a watermarked sample as non-watermarked.

prompt to trigger Yang watermark is "a rdc style cat" and the prompt to trigger Yin watermark is "a ⟨s1⟩ ⟨s2⟩ style cat".

### A.4 DISCUSSION ABOUT POTENTIAL ATTACKS.

By exploiting watermark transferability, a watermarked LoRA for diffusion models can be created by integrating the watermark LoRA with a task-specific LoRA. An adversary might attempt to strip away watermark parameters while preserving task-relevant ones. To explore this, we apply Independent Component Analysis (ICA) to decompose the integrated weights and remove the watermark component. However, as shown in Fig. 8 in Appendix, the cosine similarity distribution of the ICA components reveals significant overlap between the two LoRAs, rendering this approach ineffective.

Model stealing is another threat, where queries to the target model are used to train a surrogate. Defenses such as Entangle (Jia et al., 2021) and MEA (Lv et al., 2024) introduce robust watermarks to counter this. While these strategies can be adapted to enhance our method, this work focuses on improving watermark reliability under LoRA integration via addition and negation, rather than on resisting model extraction.

### A.5 THE USE OF LARGE LANGUAGE MODELS.

We used a large language model (ChatGPT) to improve the clarity and fluency of the manuscript text. All the ideas, analyses, and conclusions are solely those of the authors.

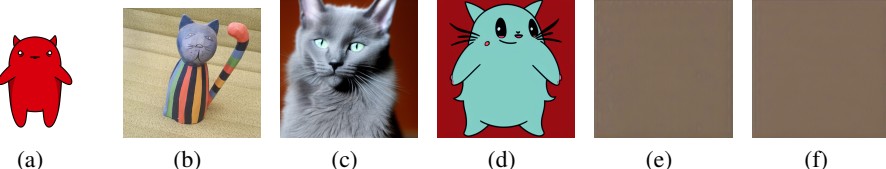

(a)      (b)      (c)      (d)      (e)      (f)

Figure 6: (a) Yin style, (b) Yang style, and main task performance and generated images before (c, d) and after (e, f) clip with Yang watermark triggered.

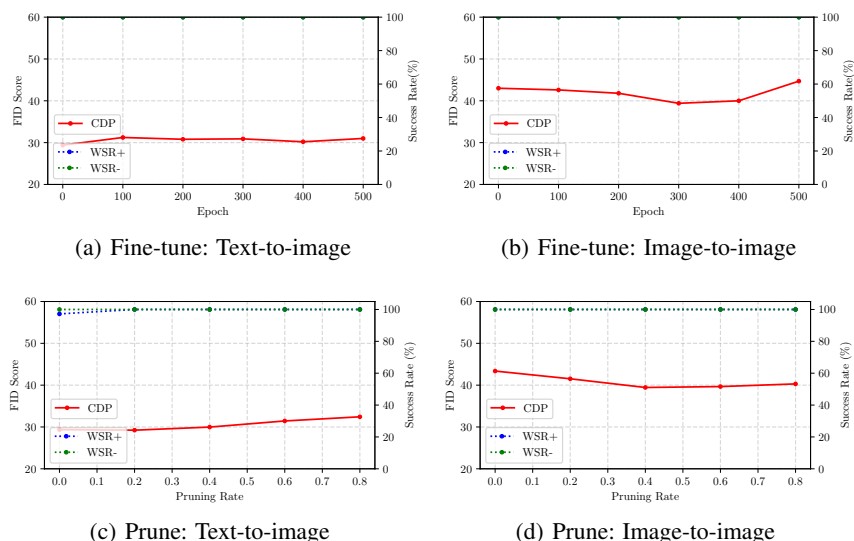

(a) Fine-tune: Text-to-image        (b) Fine-tune: Image-to-image

(c) Prune: Text-to-image        (d) Prune: Image-to-image

Figure 7: CDP and WSR as a function of retraining epoch and pruning rate on Stable Diffusion model in text-to-image and image-to-iamge tasks.

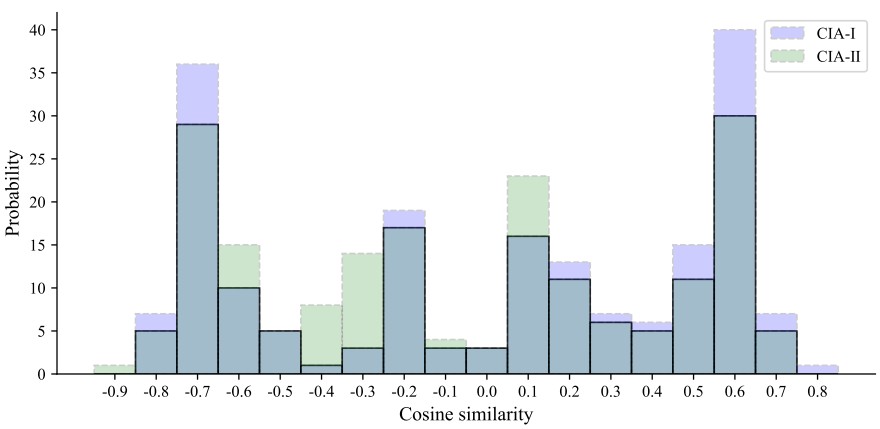

Figure 8: ICA results distribution on Stable Diffusion.

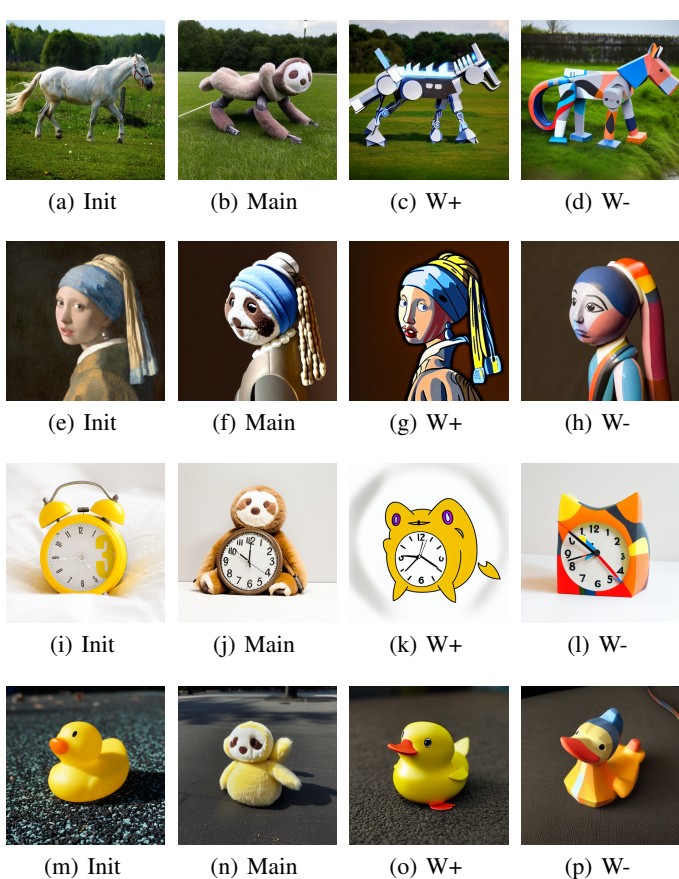

(a) Init     (b) Main     (c) W+     (d) W-

(e) Init     (f) Main     (g) W+     (h) W-

(i) Init     (j) Main     (k) W+     (l) W-

(m) Init     (n) Main     (o) W+     (p) W-

Figure 9: Watermarked LoRA on stable diffusion model in image-to-image task. The main task is "plushie slothof". Each row shows images generated by the base model, the model with the watermark LoRA applied to the main task, and the images triggered by the Yang and Yin watermarks, respectively. The prompts for each row are as follows: "style of [MASK], robotic horse with rocket launcher", "style of [MASK], a girl with pearl earring ", "style of [MASK], a clock " and "style of [MASK], a duck toy".

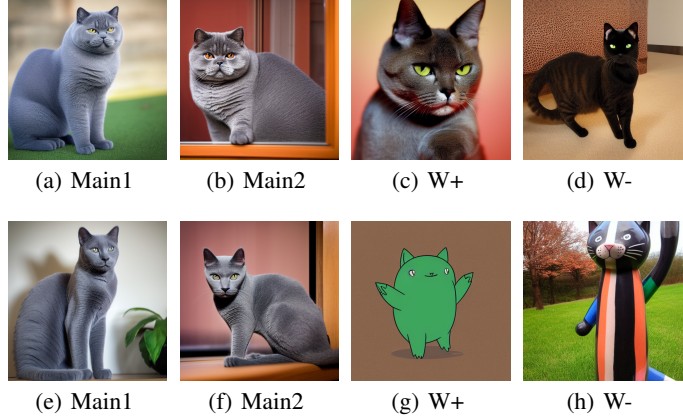

(a) Main1     (b) Main2     (c) W+     (d) W-

(e) Main1     (f) Main2     (g) W+     (h) W-

Figure 10: Clean LoRA (the first row) and watermarked LoRA (the second row) in text-to-image task.

Table 4: LoRA candidates used in the experiments on Flan-t5-large

| number | LoRA name |
| --- | --- |
| 1 | lorahub/flan_t5_large-super_glue_wic |
| 2 | lorahub/flan_t5_large-wiki_qa_Jeopardy_style |
| 3 | lorahub/flan_t5_large-newsroom |
| 4 | lorahub/flan_t5_large-wiqa_what_is_the_final_step_of_the_following_process |
| 5 | lorahub/flan_t5_large-race_high_Select_the_best_answer |
| 6 | lorahub/flan_t5_large-glue_cola |
| 7 | lorahub/flan_t5_large-word_segment |
| 8 | lorahub/flan_t5_large-wiki_qa_found_on_google |
| 9 | lorahub/flan_t5_large-anli_r1 |
| 10 | lorahub/flan_t5_large-quail_context_question_description_answer_text |
| 11 | lorahub/flan_t5_large-wiqa_what_is_the_missing_first_step |
| 12 | lorahub/flan_t5_large-imdb_reviews_plain_text |
| 13 | lorahub/flan_t5_large-drop |
| 14 | lorahub/flan_t5_large-qasc_qa_with_combined_facts_1 |
| 15 | lorahub/flan_t5_large-duorc_SelfRC_question_answering |
| 16 | lorahub/flan_t5_large-wiki_bio_comprehension |
| 17 | lorahub/flan_t5_large-adversarial_qa_dbidaf_question_context_answer |
| 18 | lorahub/flan_t5_large-quarel_choose_between |
| 19 | lorahub/flan_t5_large-wiki_bio_who |
| 20 | lorahub/flan_t5_large-adversarial_qa_droberta_tell_what_it_is |
| 21 | lorahub/flan_t5_large-lambada |
| 22 | lorahub/flan_t5_large-ropes_prompt_beginning |
| 23 | lorahub/flan_t5_large-duorc_ParaphraseRC_movie_director |
| 24 | lorahub/flan_t5_large-squad_v1.1 |
| 25 | lorahub/flan_t5_large-adversarial_qa_dbert_answer_the_following_q |

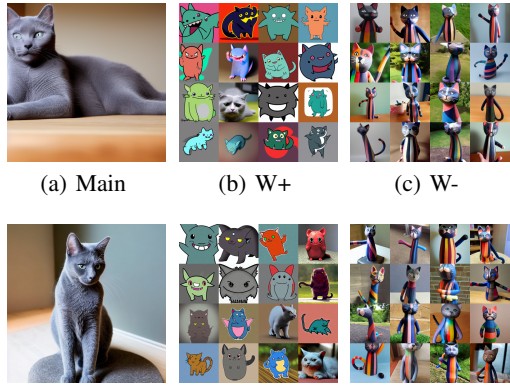

| (a) Main | (b) W+ | (c) W- |
| --- | --- | --- |

| (d) Main | (e) W+ | (f) W- |
| --- | --- | --- |

Figure 11: Watermarked LoRA in text-to-image task before (the first row) and after (the second row) ANP.

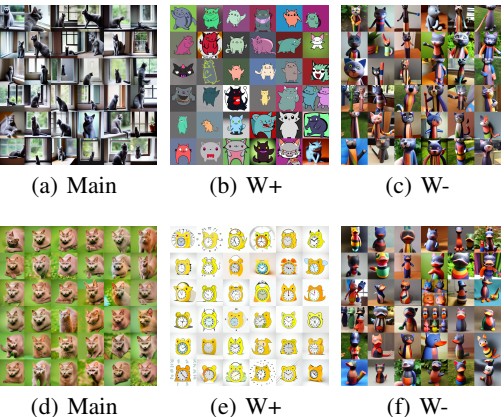

| (a) Main | (b) W+ | (c) W- |
|---|---|---|
| (d) Main | (e) W+ | (f) W- |

Figure 12: Watermarked LoRA on stable diffusion model under the fine-tuning epoch of 100. The first row is in text-to-image task and the second row is in image-to-image task.

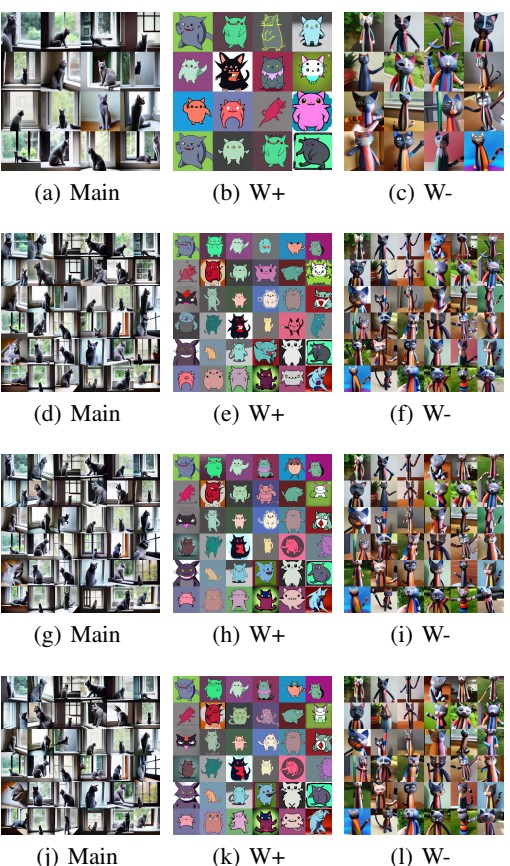

| (a) Main | (b) W+ | (c) W- |
|---|---|---|
| (d) Main | (e) W+ | (f) W- |
| (g) Main | (h) W+ | (i) W- |
| (j) Main | (k) W+ | (l) W- |

Figure 13: Watermarked LoRA on stable diffusion model in text-to-image task under the prune proportion of 0, 40%, 60%, 80%.

