# OpenReview forum: "LoRAGuard: An Effective Black-box Watermarking Approach for LoRAs"
_ICLR.cc/2026/Conference — ICLR 2026 Conference Withdrawn Submission_

### Official Review · Reviewer_qRvT · 2025-10-22

**Soundness:** 2
**Presentation:** 2
**Contribution:** 2
**Rating:** 4
**Confidence:** 2

**Summary:**

This work proposes LoRAguard, a Yin-Yang-based watermarking technique - a black-box method that embeds a watermark without requiring alterations to the model's weights.
The main advantage of this watermarking technique is that is generic and can be easily applied without forcing the model's owner to perform an intesive re-training.
The work also tests the effectiveness of their technique by implementing it on language models and diffusion models.

**Strengths:**

Watermarking is definitely an important topic, making the subject of this work relevant.
The proposed technique is lightweight and can be used without heavily modifying the weights of a pre-trained model.
This is particularly important for large models that cannot be fully retrained or altered, as it avoids wasting resources.

Strenghts:

- The proposed approach is black-box, making it easy to deploy.
- Experimental evaluation demonstrates the applicability of this approach to different model types, including language models and diffusion models.

**Weaknesses:**

- Security is purely heuristic, and no theoretical analysis of security is provided. Due to the fragility of watermarking, this makes the paper weak on this point.

- The paper does not properly explain whether the watermarking scheme requires a secret key. Some sort of secret is required to have a robust security guarantee.

- Several works have demonstrated (including from a theoretical perspective) that watermarking can be easily and generically removed. This work does not cite or compare itself against these attacks.

Other comments:

In my opinion, the paper should improve its exposition on which watermarking scenarios it is trying to address.
To my understanding, the goal is to watermark the model itself rather than the model outputs.
However, for verifying the watermark, the procedure requires submitting prompts/inputs to the target (likely in a black-box fashion) and analyzing its responses for the Yin-Yang watermark.

Besides the fact that it is unclear whether verification requires a secret key, the paper misses important citations of works that theoretically explain why watermarking schemes are fragile and can be generically broken when embedded into outputs.

For example, [Zhang et al. ] demonstrates that performing a generic random walk over the output space of a watermarked model is sufficient to generate high-quality, unwatermarked outputs.
Recently, [Francati et al.] demonstrated a similar generic attack at the bit level. Moreover, [Y] shows that crop-and-resize attacks can also effectively remove watermarks from images, aligning with the theoretical analysis.

Given these works, the paper should properly discuss how the proposed scheme may be resilient to such attacks.
In particular, a naive method leveraging [Zhang et al. ] or [Francati et al.] to generate unwatermarked outputs could be:

- Generate a watermarked answer by running the LoRAguard-based model.
- Before releasing the answer publicly, apply the procedure described in either [Zhang et al.] or [Francati et al.] to produce a similarly high-quality but unwatermarked answer.
- Release the latter output.

Based on the results of [Zhang et al. ,Francati et al.], the released outputs would likely not contain the watermark, making detection of the Yin-Yang watermark infeasible.
The paper should improve its evaluation by testing their technique against more malicious strategies (e.g., output-level attacks), including those described in [Zhang et al. ] and [Francati et al.], which also provide theoretical explanations of their inner workings.

**Questions:**

- Can you clarify how the watermark is detected? Does it require making a query to the target model and observing its output? Is the query a specific prompt?

- If detecting the watermark indeed requires observing the output, it is definitely important to test the proposed technique against the generic attacks proposed by [Francati et al.] and [Zhang et al.]. For more details, see the weaknesses section.

[Francati et al.] https://eprint.iacr.org/2025/1620
[Zhang et al.] https://proceedings.mlr.press/v235/zhang24o.html

---

### Official Review · Reviewer_Dmqx · 2025-10-27

**Soundness:** 2
**Presentation:** 2
**Contribution:** 2
**Rating:** 2
**Confidence:** 4

**Summary:**

The paper proposes LoRAGuard, a black-box watermarking approach tailored to LoRA adapters. It targets two LoRA-specific challenges that make prior backdoor-style watermarking unreliable: composition with multiple LoRAs, which dilutes backdoor effects, and the use of negation operations, which can erase backdoors or flip behaviors.

**Strengths:**

1. The Yin–Yang construction is very simple and easy follow.
2. The paper tries both LLMs and diffusion models, evaluates addition and negation, and varies key factors.

**Weaknesses:**

1. It presumes the stolen LoRA is integrated into the same base model family and that owners can query the suspect system. Cross-base or cross-version behavior is not tested.
2. It is still a backdoor watermark, which is detectable under careful forensic analysis. This watermark can be removed under heavy retraining or carefully designed purifications, and dependent on trigger rarity.
3. There’s no formal analysis of query complexity, error rates, or optimal thresholds for black-box verification, which matters for real-world deployment.
4. The employed LLM models are not sufficient. Add stronger baselines and broader experiments.
5. Provide a clear black-box verification protocol with false positive/negative rates, number of queries, and thresholds.

**Questions:**

Can you evaluate more removal/mitigation strategies relevant to LoRA (e.g., rank changes, reparameterization, quantization, cross-base integration, etc)?

---

### Official Review · Reviewer_N4E9 · 2025-10-31

**Soundness:** 1
**Presentation:** 2
**Contribution:** 1
**Rating:** 0
**Confidence:** 4

**Summary:**

In this work, the authors propose a new **model** watermarking method used to trace the use of LoRA specifically. The main idea is to convert methods for backdoor into a method to identify a LoRA in a black-box setting through queries.
The stated goal of the authors is to solve this traceability problem under two threat scenarios:

**Multi-LoRA Integration:** How to preserve the trigger effectiveness when the target LoRA is combined with other LoRAs via addition.

**Negation Operation:** How to preserve the trigger effectiveness when a negation operation is applied to the LoRA, a technique used for tasks like unlearning or domain transfer.

To solve these problems, the authors propose to inject a backdoor made out of two component:
a so called "yang" component trained to be robust when added to other "shadow LoRAs" and a so-called "yin" component trained to be robust when the negation operation is applied.

The authors evaluate their method for a single old LLM (flan-t5) and a single diffusion model (a non specified Stable-Diffusion) under different attack scenarios such as fine-tuning, pruning and backdoor detection.

**Strengths:**

Given that the paper is fundamentally flawed in its method and experimental design (see weaknesses), I am unable to assess the strengths of the paper.

For what it's worth, the author do seem to have a produced a robust and effective backdoor method ! But this is not what is advertised or tested in the paper.

**Weaknesses:**

**Lackluster experiments, low replicability**: Overall, the experimental study is insufficient. The method is evaluated on only 36 images generated by **watermarked** LoRAs and **0** (sic!) images generated by non-watermarked models. No comparison with another baseline is provided for comparison. Only one model per modality is used, and the stable diffusion model used is not even specified. Loras are trained in-house for Stable Diffusion on unspecified 10 images only. On the other hand, they are downloaded from Huggingface for Flan-T5, without giving good explanation for this choice. The author also use a subjective measure to decide on the success of their method (see after) which is highly unscientific.

**Lack of clarity in the problem specification**: The author never formally define what they try to accomplish. Only the a vague "We aim to trace the unauthorized misuse of LoRAs using watermark embedding". The reader must infer that successful tracing means that the trigger successfully activates the backdoor. This already lacks clarity and should be specified in words in the papr. Secondly, even though the watermark success can be well defined for t5-flan (is the class output correct?), it is less clear how it should be defined for images.  The authors say "A user study is conducted to assess WSR for Stable Diffusion". This means that the presence of the watermark is decided on the subjectivity of a random user, not by an objective metric (e.g., a CLIP-score, a feature-space distance, or perceptual hash), which I find highly problematic. This leads me to the most important weakness.

**Badly designed watermarking**: The authors assume success as soon as a single trigger successfully activates the backdoor on watermarked content.  This is very wrong. Since a probability of false-alarm is never  empirically quantified on non-watermarked LoRAs, we can't know if success is due to random chance or actual signal detection. The watermarking hypothesis testing problem is never formally defined, and as such, FPR and TPR metrics are naturally absent from the paper. Even worse, in the image case, the author propose no objectively measurable way to verify that the output of the trigger is what the model owner embedded.  In the current way the paper is written, it is unclear how a model owner would prove ownership using this method: since there is no counterfactual to the trigger's response (what is the expected image generated by "rdc" for a non watermarked model ?), how can i know this LoRA's response to this specific trigger actually demonstrate ownership of the model? The paper's "proof" of ownership relies entirely on the secrecy and assumed uniqueness of the (trigger, target) pair. The authors seem to miss a whole piece of the watermarking system, rendering the proposed method useless in the advertised setting.

**Questions:**

Overall, I would respectfully ask the authors to review the entire paper after designing a correct formalization for the problem they are trying to solve. In its current state, this is not a watermarking paper, and the scientificity of its experiments is highly dubious. Specifically, I would advise the authors to:

- Provide more meaningful experiments using more recent and relevant models, as well as using more samples to compute significant metrics.
- Use objective measure to match the expected target with the trigger's response (e.g., a CLIP-score, a feature-space distance, or perceptual hash)
- Formalize the watermarking problem as a hypothesis test and derive sound statistical measurement on the power and PFA of the test, in the same way as is done in all the papers the authors actually cite in the related works
- Pay close attention to the replicability of their experiment
- Why talk about content watermarking in the related works, when the authors actually only perform **model** watermarking?


In its current state, I can only recommend a strong rejection for this paper, and I doubt I could raise my score to an acceptance given how much work would be needed to match ICLR requirements.

---

### Official Review · Reviewer_HEQD · 2025-11-03

**Soundness:** 3
**Presentation:** 3
**Contribution:** 2
**Rating:** 4
**Confidence:** 4

**Summary:**

This paper addresses the problem of detecting unauthorized misuse of LoRAs by proposing LoRAGuard, a black-box watermarking technique. The method introduces two main contributions: (1) a "Yin-Yang" watermark design where the Yang watermark is triggered during addition operations and the Yin watermark during negation operations, and (2) a shadow-model-based training approach that integrates multiple unrelated LoRAs during watermark embedding to improve robustness in multi-LoRA scenarios. The authors evaluate their approach on Flan-T5-large and Stable Diffusion models, reporting nearly 100% watermark verification success rates.

**Strengths:**

1.	The paper identifies a relevant problem - tracing unauthorized LoRA misuse is indeed important as these models become more widely shared and deployed in various applications.
2.	The experimental evaluation covers multiple aspects including effectiveness under different numbers of LoRAs, various weight parameters, and robustness against fine-tuning and pruning attacks.
3.	The approach works across different model types (language models and diffusion models), demonstrating some generality.
4.	The writing is generally clear in explaining the technical approach, and the figures help illustrate the main concepts.

**Weaknesses:**

The paper tackles an interesting problem but falls short in several critical areas. The technical contribution feels incremental - essentially training two backdoors instead of one. While the shadow model training shows empirical benefits, the lack of principled justification makes it hard to understand when and why it works. The experimental evaluation, though covering multiple dimensions, remains limited in scope with only two base models tested. More concerning is the lack of comprehensive comparisons with existing watermarking methods and the incomplete treatment of adaptive adversaries. The stealthiness claims are overstated given the mixed results in Section 5.5. Details are as follows:
1.	The core contribution essentially amounts to training two separate backdoors - one with positive weights (Yang) and one with negative weights (Yin). This is a relatively straightforward extension of existing backdoor-based watermarking methods rather than a fundamental innovation. The shadow model training, while helpful, also lacks deep technical insight into why this particular approach is optimal.
2.	The evaluation is limited to only two base models (Flan-T5-large and Stable Diffusion). Given the diversity of model architectures and LoRA applications, this feels insufficient to make strong claims about the method's general effectiveness. More recent or diverse models (e.g., Llama, Mistral, SDXL variations) would strengthen the evaluation.
3.	The paper primarily compares against BadNets and mentions one prior work (Liu et al., 2024b) but doesn't provide comprehensive comparisons with other watermarking or backdoor detection methods. The lack of ablation studies to justify design choices (e.g., why dropout over other regularization techniques) is concerning.
4.	The Way2 approach of generating shadow LoRAs using random Gaussian noise based on statistics from other LoRAs seems ad-hoc. There's no theoretical or empirical justification for why this should produce meaningful shadow models. The fact that Way1 and Way2 perform similarly might actually suggest the shadow models aren't contributing much beyond regularization.
5.	The threat model assumes adversaries only use simple addition/negation operations and doesn't consider more sophisticated attacks like adaptive adversaries who know about the watermarking scheme, or more complex LoRA composition methods. The fine-tuning defense evaluation uses only 1,500 samples, which seems limited.
6.	The stealthiness experiments (Section 5.5) show mixed results but are presented somewhat optimistically. For instance, the RAP and ONION results show that detection is possible with reasonable FRR/FAR tradeoffs, which contradicts claims of strong stealthiness.

**Questions:**

1.	How does the method perform on more recent and diverse model architectures? Have you tested on models like Llama-2/3, Mistral, etc?
2.	Can you provide theoretical justification or more detailed ablation studies on why the shadow model approach with dropout works? What happens if you use other regularization techniques?
3.	How does computational cost scale with the number of shadow models used during training? What's the overhead compared to standard LoRA training?
4.	Have you considered adaptive attacks where adversaries know about your watermarking scheme and specifically try to remove the Yin-Yang watermark?
5.	The transferability claim in Section 4.4 is interesting but underexplored. Can you provide more extensive evaluation of watermark transfer to other task-specific LoRAs?

---

### Note · Authors · 2026-01-06

**Comment:**

We would like to withdraw this submission. Thank you for your time and consideration.

**Withdrawal Confirmation:**

I have read and agree with the venue's withdrawal policy on behalf of myself and my co-authors.